# Unraveling the Genetic Diversity and Population Structure of Bangladeshi Indigenous Cattle Populations Using 50K SNP Markers

**DOI:** 10.3390/ani11082381

**Published:** 2021-08-12

**Authors:** Mohammad Shamsul Alam Bhuiyan, Soo-Hyun Lee, Sheikh Mohammad Jahangir Hossain, Gautam Kumar Deb, Most Farhana Afroz, Seung-Hwan Lee, Abul Kashem Fazlul Haque Bhuiyan

**Affiliations:** 1Department of Animal Breeding and Genetics, Bangladesh Agricultural University, Mymensingh 2202, Bangladesh; msabhuiyan.abg@bau.edu.bd; 2Division of Animal Breeding and Genetics, National Institute of Animal Science, Cheonan 31000, Korea; lhyungm@korea.kr; 3Biotechnology Division, Bangladesh Livestock Research Institute, Savar 1341, Bangladesh; smjhossainblri@yahoo.com (S.M.J.H.); debgk2003@yahoo.com (G.K.D.); famukta@yahoo.com (M.F.A.); 4Division of Animal and Dairy Science, Chungnam National University, Daejeon 34134, Korea

**Keywords:** genetic diversity, population structure, indigenous cattle, SNP array, Bangladesh

## Abstract

**Simple Summary:**

Indigenous cattle have extraordinary adaptation capability to diverse environments under low input production system. However, the population size is declining rapidly in Bangladesh due to massive imports of high yielding dairy breeds. The genetic diversity measures are important for assessing population architecture as well as for development of conservation strategies. The aim of this study was to investigate genetic variability and population structure of indigenous cattle genetic resources of Bangladesh using Illumina Bovine SNP50K BeadChip genotyped data. Similar to other zebu populations, low genetic diversity measures were found in Bangladeshi cattle populations. Our findings revealed their distinct genetic structure but showed low levels of genetic differentiation among the six indigenous cattle populations. Moreover, admixture and phylogenetic analysis highlighted historical gene flow among the studied populations. Altogether, our findings provide a comprehensive genomic information on indigenous cattle populations of Bangladesh that could be utilized in their future conservation and breeding research.

**Abstract:**

Understanding the genetic basis of locally adapted indigenous cattle populations is essential to design appropriate strategies and programs for their genetic improvement and conservation. Here, we report genetic diversity measures, population differentiation, and structure of 218 animals sampled from six indicine cattle populations of Bangladesh. Animals were genotyped with Illumina Bovine SNP50K BeadChip along with genotyped data of 505 individuals included from 19 zebu and taurine breeds worldwide. The principal component analysis (PCA) showed clear geographic separation between taurine and indicine lineages where Bangladeshi indigenous cattle clustered with South Asian zebu populations. However, overlapped clusters in PCA, heterozygosity estimates, and Neighbor-Joining phylogenetic tree analysis revealed weak genetic differentiation among the indigenous cattle populations of Bangladesh. The admixture analysis at K = 5 and 9 suggests distinct genetic structure of the studied populations along with 1 to 4% of taurine ancestry. The effective population size suggested a limited pool of ancestors particularly for Sahiwal and North Bengal Grey cattle. In conclusion, these findings shed insights into the genetic architecture of six indigenous cattle populations of Bangladesh for the first time and suggested as distinct gene pools without potential admixture with zebu or taurine populations.

## 1. Introduction

Indigenous cattle are an important livestock species in smallholder farming system of Bangladesh. They have potential contributions in rural livelihoods, nutrition security, organic agriculture, and socio-cultural and historic events. Moreover, rural farmers prefer indigenous cattle due to their better reproducibility and adaptation under low-input management practices [1,2]. In Bangladesh, the total heads of cattle are estimated to be 24.39 million [3]. Indigenous cattle are primarily categorized into five different varieties or types namely Red Chittagong (RCC), Pabna (PC), Munshiganj (MC), North Bengal Grey (NBG), and Non-descript Deshi (DES), those possess distinct coat color as well as have differences in their productivity and morphometric features [2]. These varieties have been developed in their breeding tracts due to farmers’ selection over the years particularly for milk production and phenotypes. In particular, milk production is comparatively higher in RCC, MC, and PC than NBG and DES [1,4]. In addition, Sahiwal (SL) breed were introduced in Bangladesh six decades ago as an improved zebu dairy cattle and is now sparsely distributed throughout the country.

In Bangladesh, the indigenous cattle alone predominated until the late seventies of the twentieth century. Upgradation of indigenous cattle varieties has been practiced for the last five decades to boost up milk production through the introduction of high yielding temperate and tropical dairy breeds [5]. However, indiscriminate upgrading and crossbreeding programs without proper conservation efforts have led to the production of upgraded animals’ exponentially at the expense of indigenous cattle varieties of Bangladesh [2,6]. Thereby, the population size of indigenous varieties declined rapidly, whereas MC and PC are at risk of extinction now [1]. Considering the above stated scenarios, ex situ conservation programs on PC, MC, and RCC are on-going at different government institutional herds in limited scales for more than a decade. More importantly, comprehensive information of current genetic diversity and demographic process of those indigenous populations are essential to take the necessary steps for development of sustainable breeding programs and maintenance of unique gene variants.

Genetic diversity illustrates the key aspects of differentiation among the individuals of a population that exists either at phenotypic or DNA level [7]. The measures of genetic diversity provide essential insight information on livestock populations conservation and improvement strategies, as well as their adaptation to certain environments [8]. The landscape and depth of bovine genomic research has got a new dimension with the advent of SNP chip data. Genome wide SNP50K chip have been implemented to investigate genetic diversity, population structure, level of inbreeding, admixture analysis, effective population size, linkage disequilibrium, migration events, genome-wide association studies, and detection of selection signatures in different cattle populations around the world [9,10,11,12]. Estimation of genomic breeding value and its prediction accuracy is another frontline avenue in the genetic evaluation process using SNP chip data [13]. On the other hand, more accurate diversity parameter estimates have become possible now through utilization of genomic information compared to traditional pedigree data. However, the indigenous cattle population of Bangladesh remained poorly studied especially at the molecular level. Earlier genetic characterization studies were carried out in RCC population using microsatellite markers [14] and mitochondrial DNA [15]. Analysis of SNP80 K indicine commercial chip data revealed weak genetic differentiation between RCC and DES cattle [16]. It is noted that the comprehensive genetic diversity study utilizing all indigenous cattle genetic resources of Bangladesh has not yet been performed using high density SNP markers. Therefore, the present study aimed to investigate the extent of genetic diversity, population structure, and differentiation among the six indigenous cattle varieties of Bangladesh in comparison with other reference populations or breeds.

## 2. Materials and Methods

### 2.1. Ethics Approval

For this study, blood samples were collected from institutional, private, and farmers’ herds under the approval of Ethical Committee of Bangladesh Agricultural University, Bangladesh (no. 1218/BAURES/2020/ESRC/AH/10).

### 2.2. Animal Sampling and DNA Extraction

A total of 240 blood samples were collected from six zebu cattle populations of Bangladesh (Appendix A). Among them, five indigenous cattle varieties namely RCC (*n* = 89), MC (*n* = 26), PC (*n* = 45), NBG (*n* = 21), and DES (*n* = 22) are distributed in different agro-ecological regions, while SL (*n* = 15) breed are being predominantly available in milk potential areas. Blood sampling was performed from government or institutional herds (Central Cattle Breeding and Dairy Farm, Bangladesh Livestock Research Institute, Savar, Dhaka), university managed herd (Bangladesh Agricultural University, Mymensingh, Bangladesh), private dairy farm (American Dairy Limited, Gazipur, Bangladesh), as well as from 37 smallholders’ farms (Appendix A). Precautions were taken to avoid sampling from the related individuals. Genotype of the sampled individuals was ascertained through pedigree analysis as well as in-depth enquiry of the animal owners. Blood samples were collected from jugular vein using vacutainer containing EDTA as anticoagulant and were immediately transferred to the laboratory for DNA extraction. Genomic DNA was extracted from the whole blood using the Prime Prep^TM^ DNA isolation kit (GeNet Bio Co. Ltd., Daejeon, Korea). The concentration and purity of isolated DNA were measured by the Nanodrop spectrophotometer (Model ND2000, Thermo Fisher Scientific, Wilmington, DE, USA) prior to genotyping.

### 2.3. SNP Genotyping and Quality Control

DNA samples were genotyped using Illumina bovine SNP50 v.3 BeadChip with the help of the commercial genotyping service provider (TNT Research Co. Ltd., Seoul, Korea). The cattle SNP50 chip possesses 53,218 SNPs that uniformly span over the entire bovine genome. Genome Studio^®^ software (Illumina, San Diego, CA, USA) plugin PLINK v.1.9 was employed for genotypes calling. SNP filtering was performed using PLINK v.1.9 [17] based on the following exclusion criteria: Minor allele frequency (MAF) <0.01 and call rate <0.90. SNP filtering based on the Hardy–Weinberg equilibrium (HWE) was not performed since we expected HWE deviations in some of the studied populations due to their small and possibly sub-structured population and genetic drift. Furthermore, SNPs assigned to sex chromosomes and those lacking genomic locations were excluded from the analysis. Apart from this, genotyping data of Korean Hanwoo (KPN), Chikso (CHK), and Jeju Black (JB) were made available by Dr. Seung Hwan Lee, Chungnam National University, Daejeon, South Korea. In addition, Central Asian cattle breeds Yianbian (YBH) and Mongol (MG) data were downloaded from dryad.org [18]. Genotype information of African taurine (N’Dama, ND and Oulmes Zaer, OUL) and Zebu cattle (Zebu Madagascar, ZMA and Sheko, SHK), and European taurine breeds Angus (ANG), Brown Swiss (BSW), Hereford (HFD), Holstein (HOL), Limousin (LMS), Guernsey (GNS), Santa Gertrudis (SGT), and Beefmaster (BMA) were used from the Bovine Hapmap consortium. Details on breeds or cattle populations and number of samples used in this study are presented in Appendix A. The final dataset included 723 individuals from 25 cattle populations/breeds and 35,964 SNPs in the merged data file.

### 2.4. Statistical Analysis

#### 2.4.1. Genetic Diversity

To assess the within population genetic diversity, the proportion of polymorphic loci (P_N_), observed (Ho) and expected (He) heterozygosity were estimated using R package hierfstat [19]. The distribution of minor allele frequency (MAF) was grouped into six different categories based on their frequency as rare alleles (0 < maf ≤ 0.05), intermediate alleles (0.05 < maf ≤ 0.10), and common alleles (0.10 < maf ≤ 0.50). Diversity indices were calculated from 45,861 filtered SNPs for six zebu cattle populations of Bangladesh.

#### 2.4.2. Population Structure and Genetic Differentiation

The following approaches were employed to investigate the population structure among the indicine cattle populations of Bangladesh, as well as to assess their relationships with other cattle breeds distributed globally, principal component analysis (PCA), admixture analysis, pairwise genetic differentiation (F*_ST_*), and maximum likelihood phylogenetic tree construction. Two different datasets were used in PCA and admixture analysis, 17,477 SNPs that passed the quality control threshold from 723 individuals belong to 25 global cattle populations (Appendix A), while 18,481 SNPs were used for 264 individuals of eight zebu cattle populations. To perform PCA, PLINK v.1.9 was used to generate eigenvectors and eigenvalues, and the outputs were visualized using the R package SNPRelate [20] that demonstrate the relationships among the PC1, PC2, and PC3 coordinates. Furthermore, the population genetic structure assessment was performed using ADMIXTURE v.1.3 software [21], assuming a number of hypothetical population clusters (K) ranging from 2 to 9 and the output was visualized using R plots. The optimum number of K value was obtained from the lowest cross-validation (CV) error estimation. The pairwise F*_ST_* and Nei genetic distances among the populations were calculated using the R package StAMPP [22]. The Neighbor Joining (NJ) tree was constructed using the SNPhylo pipeline [23] to know the evolutionary relationships and was illustrated using FigTree v.1.4.4 (http://tree.bio.ed.ac.uk/software/figtree/, accessed on 7 April 2020).

#### 2.4.3. Linkage Disequilibrium and Effective Population Size

Linkage disequilibrium (LD) was used to examine the recombination events of linked SNPs in each population and was measured as the correlation coefficient (r^2^) between two loci. SNPs spanning from 0 to 500 Kb distance were included in this step. The effective population size (Ne) was calculated at different generations from the resulted LD value according to the equation suggested by Sved [24]. The estimates of LD and Ne were performed using PLINK v.1.9 and SNeP v.1.1 [25] with default parameters, respectively. The Ne estimates were plotted over the last 0 and 10,000 generations using R software v.3.3.1 (R Foundation for Statistical Computing, Vienna, Austria) to investigate the diversity trends.

## 3. Results

### 3.1. Intra-Population Genetic Diversity

After quality control procedures, 218 samples and 35,960 SNPs were remained in the final dataset for downstream analysis of Bangladeshi cattle populations. Twenty-two animals were excluded from the analysis due to low call rate (<0.90). The genetic diversity measures are given in Table 1. Most of the Bangladeshi indigenous cattle manifested a high proportion of polymorphic loci (P_N_), varying from 0.668 in SL to 0.905 in RCC. The highest proportion of SNPs that belong to MAF category of ≤0.05 was found in RCC (0.272), whereas the lowest proportion was shown in Sahiwal breed (0.120), with overall mean of 0.197 across populations. The highest observed heterozygosity was observed in RCC (0.250 ± 0.180), while the lowest was in PC (0.211 ± 0.166), indicating higher diversity in RCC compared with other indigenous cattle populations. However, small differences were observed regarding heterozygosity measures (Ho and He) among the studied cattle populations. Minor allele frequency distributions in six oindigenous populations under study based on different categories are shown in Figure 1. The percentage of fixed SNPs (MAF = 0.00) ranged between 16.6 and 31.5% in RCC and SL cattle, respectively, with an average of 24.2% across populations. Distribution of minor allele under low frequency category (0, ≤0.05) was significantly higher in all cattle populations except SL, also depicted from Table 1. However, the proportion of MAF did not differ largely among the cattle populations for the remaining category of [0.05, 0.1], [0.1, 0.2], [0.2, 0.3], [0.3, 0.4], and [0.4, 0.5].

### 3.2. Population Structure and Genetic Differentiation

The genetic structure among the indigenous cattle population of Bangladesh and 19 selected cattle breeds across the world were assessed through PCA using 17,477 SNPs. Cattle populations were separated at sub-species and region level (Figure 2). The PC1 explained 8.96% of the total variance that clearly separated taurine breeds from the indicine breeds/populations. Among the investigated indicine breeds or populations three major clusters were observed, the first one concatenating all Bangladeshi cattle populations, while the second and third clusters enclosing South Asian (NEL and BRM) and African (ZMA and SHK) indicine cattle, respectively (Figure 2A). The PC2 accounted for 3.06% of the variation, segregated European taurine from East and Central Asian taurine breeds. In addition, African taurine (ND and OUR) occupied an intermediate position in between European dairy (BSW, HFD, HOL, and GNS) and beef (BMA and SGT) breeds (Figure 2A and Appendix A). More detailed analyses using the indigenous cattle population of Bangladesh, PC1 and PC2 illustrated only 2.66 and 1.43% of the total variations, respectively. All individuals dispersedly distributed without formation of any specific cluster. However, some individuals of RCC clearly isolated from all other Bangladeshi populations, as depicted by PC2 (Figure 2B and Appendix A).

The heat map shows Nei genetic distance and pairwise F*_ST_* values in upper and lower diagonals, respectively among the 25 cattle breeds or populations (Figure 3). The lowest genetic differentiation was observed in Bangladeshi populations that ranged between 0.00 to 0.03 and also showed similarity with South Asian (BRM and NEL) and African (SHK and ZMA) zebu cattle breeds. The lowest F*_ST_* values among the zebu cattle populations reflected their close relationship to each other. As expected, the highest differentiation (pairwise F*_ST_* = 0.16 and Nei F*_ST_* = 0.13) was obtained between Bangladeshi indigenous cattle populations and taurine cattle breeds of Africa (ND and OUL), East Asia (KPN, CHK and JB), and Europe (ANG, BSW, HFD, HOL, GNS, and LMS). However, Central Asian (YBH and MG) and European beef breeds (BMA and SGT) showed moderate genetic differentiation (pairwise F*_ST_* = 0.08 to 0.11 and Nei F*_ST_* = 0.08 to 0.14) with respect to Bangladeshi indicine populations.

The admixture analysis inferred clustering patterns among Asian zebu cattle populations based on shared ancestry (at K = 3) that separates Bangladeshi Indigenous cattle from NEL cattle (Figure 4). The optimal K-value was acquired (K = 4) based on the lowest cross-validation error (Appendix A) that highlighted SL cattle shared major ancestry with Bangladeshi populations, where NEL and BRM clearly separated from them. On average, the Bangladeshi cattle had 56.0 to 86.0% similar genetic background with SL but only around 1.0 to 5.0% similarity was observed with NEL and BRM cattle (Table 2). More specifically, RCC populations demonstrated two distinct genetic backgrounds where 61% was similar with South Asian zebu cattle and the remaining 39% was accounted for population specific. Unexpectedly, 1–4% taurine ancestry (Table 2) identified between HOL and indigenous cattle population of Bangladesh might be *inter se* breeding. When another 11 European dairy and beef breeds (ANG, BMA, BSW, CHA, GNS, HFD, HOL, JER, LMS, PMT, and SGT) each possess 17,477 SNPs included with Bangladeshi indigenous cattle populations, K = 9 was found as the optimum K value (Appendix A). The admixture analysis at K = 2 grossly separated Bangladeshi indigenous cattle populations to European cattle breeds. At K = 5 and 9 each breed created their own cluster where all Bangladeshi cattle populations formed a separate cluster having minimum admixture of taurine dairy breeds (Appendix A). However, four beef breeds (BMA, LMS, PMT, and SGT) showed a certain level of zebu inheritance, while three of them (CHA, LMS, and PMT) manifested almost a similar genetic background. The degree of admixture declined with the increment of ancestry number through inclusion of 17 European cattle breeds to the present dataset. The maximum likelihood based phylogenetic tree showed evolutionary relationships among the Bangladeshi indigenous cattle, starting with the DES cattle without forming any specific cluster (Figure 5). However, some sub-clusters were noticed in the case of RCC, PC, and SL populations. Moreover, this illustration suggests their weak differentiation along with the historical exchange of genetic materials among the studied populations.

### 3.3. Analyses of Linkage Disequilibrium and Effective Population Size

The extent of LD was assessed up to 500 kb using pairwise r^2^ for six zebu cattle populations separately. The r^2^ values both at shorter and longer genetic distances varied among the cattle populations. The most rapid LD decaying pattern was observed in shorter distances of up to about 30 kb (Figure 6A). In general, levels of pairwise LD across the genome dropped with the advancing distance between adjacent SNPs. SL cattle displayed the higher LD value across the genomic distance, while RCC had the lowest LD value. The LD values (average r^2^ in 200 to 500 Kb fragments) were 0.12, 0.08, 0.09, 0.07, 0.06, and 0.04 for SL, DES, NBG, PC, MC, and RCC, respectively. The effective population size was evaluated based on LD estimates (r^2^) from the recent generation to 10,000 generations ago (Figure 6B). All six cattle populations showed sharp declining trends in their Ne values over time. Among the populations, Ne ranged from 494 to 2058 animals 100 generations ago. However, in the recent past (until five generations ago), RCC had the highest Ne (108.29), while the lowest value (26.02) was found in SL and the intermediate estimates were found to be 47.68, 40.87, 33.69, and 46.64 in MC, DES, NBG, and PC, respectively, representing a narrow genetic pool in the studied populations.

## 4. Discussion

Indigenous cattle populations are mostly random bred and non-selected genetic resources that harbor unique gene pools resulting from their adaptation to the local environment [13]. Hence, more diverse alleles are expected in those populations due to the absence of intense directional selection for production traits. In this study, genome-wide SNP data were analyzed to advance our knowledge on genetic architecture and diversity of Bangladeshi indigenous cattle in the worldwide population context.

The results of this study depicted relatively low genetic diversity measures in terms of P_N_, Ho, and He in Bangladeshi cattle populations and are in agreement with the previous reports of Xu et al. [12], Mustafa et al. [26], and Zhang et al. [27]. Previous studies also reported relatively higher diversity measures in taurine breeds compared to their indicine counterparts using the same genotyping array [9,11,13]. In fact, the low representation of indicine breeds in the SNP genotyping array results in the ascertainment bias towards taurine breeds that disproportionate MAF distribution among the Asian and African indicine breeds or subpopulations [11,28]. In addition, SNP filtering from combined datasets of 25 cattle breeds/populations kept a low number of SNPs to be investigated. However, the SNP genotyping and quality control methods are comparable with the previous studies in different indicine cattle populations where minimum bias on their results have been reported [26,29]. Similar to the present findings, the average minor allele frequency (MAF) ranging from 0.11 to 0.23 that was observed in three South Asian zebu breeds (Gir, Sahiwal, and Nellore), 10 indicine breeds of Pakistan, and two Chinese indicine cattle Wenshan and Nandan [12,26,27]. In earlier studies, Uzzaman et al. [16] and Edea et al. [29] found a bit higher MAF (0.28 to 0.31) in two Bangladeshi (RCC and DES) and three Ethiopian (Begait, Guraghe, and Ogaden) zebu populations. In our study, the MAF distribution pattern was consistent with Chagunda et al. [11] but contradicted with the reports of Pérez O’Brien et al. [30] and Bejarano et al. [31] who found a higher proportion of common alleles (≥0.10) in taurine cattle as compared to indicine breeds. The higher percentage of low MAF associated with greater genetic diversity, affects LD distribution and extends within the population [30,32], has larger effects and better genomic predictive ability for quantitative traits in cattle [31], and supports the present findings.

On the other hand, the estimated heterozygosity values of this study (average Ho = 0.22 ± 0.17 and He = 0.19 ± 0.12) were in agreement with most of the previous studies involving different indicine populations. Sharma et al. [9] reported Ho and He levels in Brahman, Nellore, and Gir breeds ranging from 0.20 to 0.22 and 0.15 to 0.18, respectively. The current results are also consistent with those reported by Zhang et al. [27] and Chagunda et al. [11] in Asian (Gir, Sahiwal) and African (East African Shorthorn Zebu) zebu cattle that ranged between 0.22 and 0.27. However, worldwide distributed taurine breeds presented relatively higher MAF (ranging from 0.23 to 0.31), P_N_ (varies from 0.80 to 0.97), Ho (0.30 and 0.39), and He (0.24 to 0.38) compared to the present findings as reported by Sharma et al. [9], Kim et al. [10], and Upadhyay et al. [13].

Several statistical approaches have been utilized to infer relationships among the cattle populations as well as to assign individuals in their respective populations. The principal component analysis (PCA) using the genotyped dataset of this study along with 19 worldwide distributed cattle populations’ data revealed that Bangladeshi indicine cattle clustered with South Asian zebu cattle and a bit away from African zebu populations. Earlier SNP-based studies in cattle and sheep demonstrated that first (PC1) and second (PC2) components of PCA clustered breeds or populations were based on their geographic origin [33]. Likewise, our PCA analysis demonstrated geographic origin and sub-species oriented clustering that depict the recent division of Bangladeshi cattle from their primary center of zebu domestication, the Indus Valley and is in accordance with the findings of Chen et al. [34]. The major influence of South Asian zebu to Bangladeshi indigenous cattle was also ascertained through the genetic composition analysis (Table 2), as well as from the cattle breeding history of Bangladesh. During the British colonial period, Indian zebu male mediated introgression had been employed to improve the small sized indigenous cattle of Bangladesh [15] that also supports the present SNP data-based findings.

In our study, the clustering pattern in PCA (Figure 2B), pairwise F*_ST_* and Nei genetic distance (Figure 3), and population STRUCTURE analysis (Figure 4) indicate weak differentiation among Bangladeshi cattle populations. The absence of well-defined but overlapped clusters in PCA is probably due to the historical gene flow among themselves. In fact, the indigenous cattle population was a large random bred population over the centuries. They have been categorized in the recent past based on coat color and morphometric features, where sufficient genetic differentiation has not yet been established through within population selective breeding [1,14]. However, one third of individuals belong to RCC clearly separated from other zebu populations that reveals a genetically distinct population within RCC. This finding is supported by the mtDNA based study of Bhuiyan et al. [15] who found two separate clusters in RCC population that might be due to ex situ conservation through closed herd nucleus breeding and thus created genetic drift in the isolated population. It is noted that blood sampling was performed both from in and ex situ individuals for the said populations. Similar to our findings, Ethiopian cattle populations had low F*_ST_* values of 0.011 to 0.012 [35] using microsatellite markers and low-density taurine derived chip. However, higher F*_ST_* values were observed between Gir and EASZ (0.11) breeds [11], Ethiopian and Asian zebu (0.07) populations [29], between Deoni and Ongole (0.12) breeds of India [36]. In addition, Shah et al. [37] reported the low F*_ST_* estimate between Kankrej and Malvi cattle breeds of India (0.013) and is similar to our results. Earlier F*_ST_* estimates by Sharma et al. [9] observed genetic closeness among Asian zebu breeds compared to European, African zebu, and taurine cattle and supports our findings. Taken together, the lowest genetic distance observed among the Bangladeshi indigenous cattle populations suggest that they did not differentiate well as an independent breed and might be due to their common ancestral origin and exchange of genetic materials in the recent past [29]. This phenomenon is also supported from the phylogenetic results (Figure 6), where the absence of population specific clusters suggests strong gene flow among each other.

Similar to the PCA, consistent results were also obtained from the admixture analysis. Among the zebu populations, the BRM and NEL clearly separated from six Bangladeshi cattle starting at K = 3. The distinct genetic composition of BRM and NEL might be their geographic isolation and long-term selection for desired traits [18]. However, a small proportion of zebu admixture in the indigenous cattle variety of Bangladesh probably is due to their common ancestral origin. When SNP information of worldwide distributed cattle was included to present the dataset, clear divergence was noticed between Asian zebu (Bos indicus) and taurine (Bos taurus) cattle at all K values starting from K = 2 to 9. Our findings are in agreement with previous studies that support two independent domestication events that took place for two bovine species [18,38]. Moreover, taurine beef breeds mostly genetically admixed those possessed both indicine and taurine ancestry and is consistent with the findings of Decker et al. [18] and Zhang et al. [27]. However, the very low level of taurine admixture in Bangladeshi cattle population could be explained from their breeding history, where remnants of taurine introgression through indiscriminate breeding still exists.

Based on the average r^2^ value, a slower LD decay across the distance in SL cattle indicates its heterozygosity deficit compared to other five zebu populations. The lowest LD in RCC reflects a more diverse population with weak directional selection. In addition, the sharp decline of LD in short distance (Figure 6A) is also an indication of high haplotype diversity in the studied populations. In our study, LD values decrease in a constant manner as the distance among markers increase and the similar trend was reported in cattle by Kim et al. [10] and Shin et al. [39]. Usually, LD characteristics are population or breed specific despite the fact that several factors are associated with LD patterns and scale within or between populations such as demographic history and population structure, sample size, marker type and density, MAF thresholds, selection and method of LD measurement [40]. The estimated effective population size (Ne) was relatively low in the studied population except RCC. However, our results are comparable with the findings of Zhang et al. [27] who found that Ne values varied from 10 to 259 in 17 Chinese indigenous cattle breeds. In addition, Ne of three Korean cattle populations (BH, BRH, and JB) until 13 generations ago were 83, 59, and 67, respectively [9] and agrees with this study, but higher Ne values as 260, 202, and 55 were reported by Kim et al. [10] for those aforementioned genotypes. Importantly, the methodological aspects could potentially affect Ne estimates from the LD pattern, particularly LD from r^2^ estimates. For instance, a small sample size leads to bias LD estimation and therefore, a minimum sample size of 55 was suggested for accurate LD estimation based on r^2^ values [41]. The lower Ne values of the present study might be associated with the relatively small sample size and low number of filtered SNPs included in the analysis. Altogether, LD patterns and Ne reflect various demographic and selection events, as well as bear significance to understand the genomic architecture of a breed or population [42].

## 5. Conclusions

This study provided an important glimpse of genetic diversity, population differentiation, and structure among indigenous cattle of Bangladesh for the first time using high density genome-wide SNP data. In this study, we found relatively low genetic diversity measures that were comparable to other zebu populations worldwide. Furthermore, Bangladeshi cattle population did not differentiate well as independent breeds, revealed by their low F*_ST_* values. A certain amount of zebu and taurine admixture in the investigated populations might be due to their common ancestral origin and the cattle breeding history of Bangladesh. The lack of well-defined and overlapped clusters in PCA and phylogenetic tree highlight the historical gene flow among themselves and the absence of strong directional selection over the generations. Taken together, our findings reported herein the basic features of genomic architecture of Bangladeshi indigenous cattle populations that will aid in their future conservation and genetic improvement strategies and programs.

## Figures and Tables

**Figure 1 animals-11-02381-f001:**
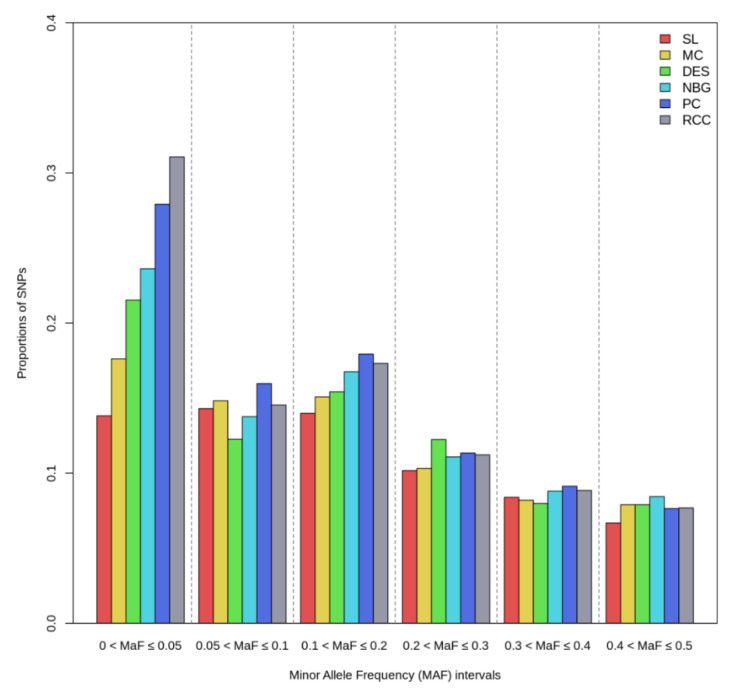
Minor allele frequency distribution across six Bangladeshi indigenous cattle populations. The included populations are Sahiwal (SL), Munshiganj (MC), Non-descript Deshi (DES), North Bengal Grey (NBG), Pabna (PC), and Red Chittagong (RCC).

**Figure 2 animals-11-02381-f002:**
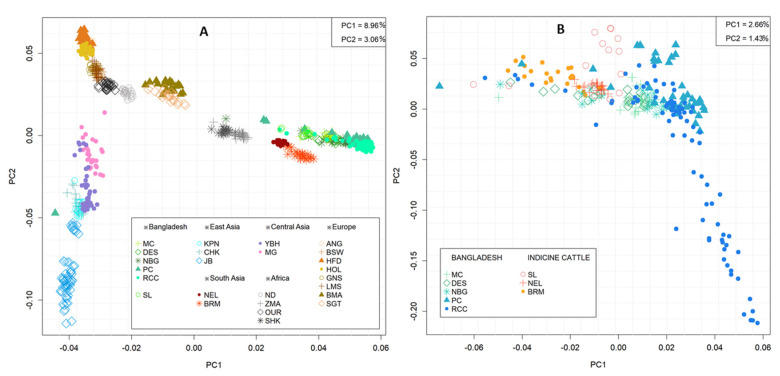
Principal component analysis of the first two axis in 723 samples from 25 cattle populations distributed worldwide (**A**) and 264 samples from six Bangladeshi Indigenous cattle along with two other indicine breeds (**B**). Cattle populations are labelled as Sahiwal (SL), Munshiganj (MC), Non-descript Deshi (DES), North Bengal Grey (NBG), Pabna (PC), Red Chittagong (RCC), Brahman (BRM), Nellore (NEL), Korean Hanwoo (KPN), Korean Chikso (CHK), Korean Jeju Black (JB), Yianbian (YBH), Mongolian (MG), Angus (ANG), Brown Swiss (BSW), Hereford (HFD), Holstein (HOL), Guernsey (GNS), Limousine (LMS), Santa Gertrudis (SGT), Beefmaster (BMA), N’Dama (ND), Oulmes Zaer (OUR), Zebu Madagascar (ZMA), and Sheko (SHK).

**Figure 3 animals-11-02381-f003:**
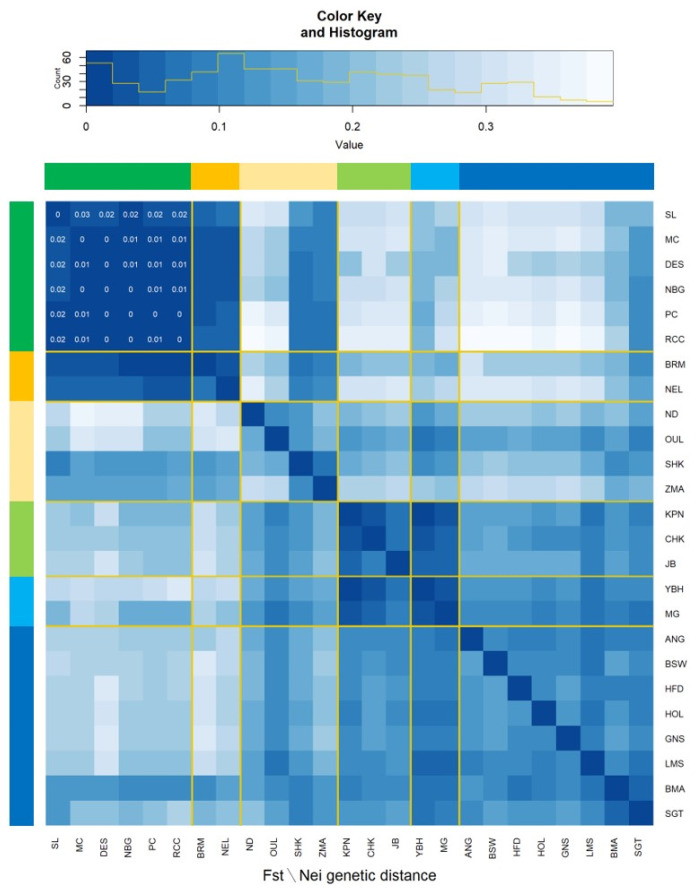
Heat map shows pairwise F_ST_ (below diagonal) and Nei genetic distance (above diagonal) values for six Bangladeshi cattle populations (green), Indian zebu cattle (orange), African cattle populations (accent gold), Korean cattle (light green), Chinese cattle populations (light blue), and European cattle breeds (blue). The map was constructed utilizing 45,861 SNPs. Cattle breeds/populations abbreviations as in Figure 2. Pairwise F_ST_ and Nei genetic distance values among six Bangladeshi populations are shown in the heat map.

**Figure 4 animals-11-02381-f004:**
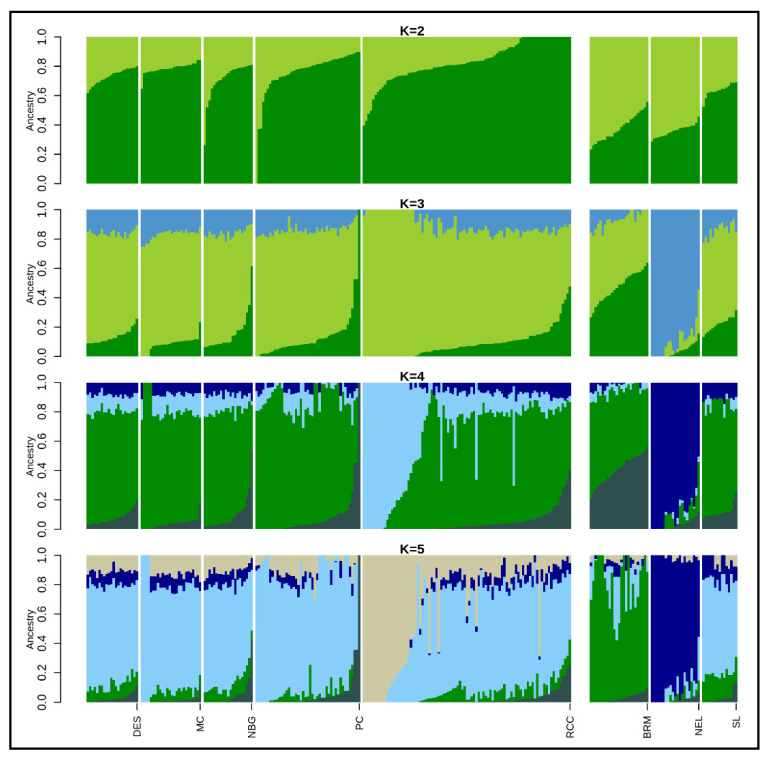
Population structure plots showing proportions of assumed ancestries in indicine cattle for K = 2–5. DES: Non-descript Deshi; MC: Munshiganj; NBG: North Bengal Grey; PC: Pabna; RCC: Red Chittagong; BRM: Brahman; NEL: Nellore; and SL: Sahiwal.

**Figure 5 animals-11-02381-f005:**
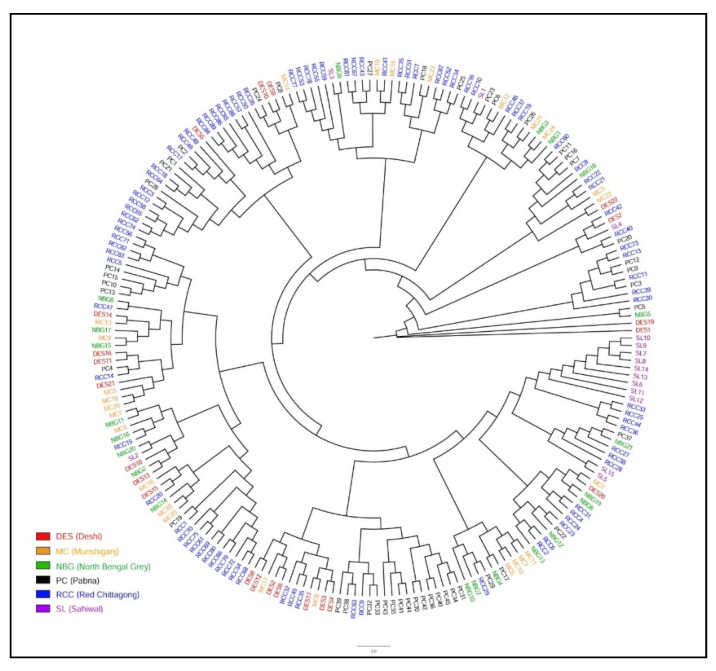
Maximum likelihood phylogenetic tree using six Bangladeshi cattle populations are Sahiwal (SL), Munshiganj (MC), Non-descript Deshi (DES), North Bengal Grey (NBG), Pabna (PC), and Red Chittagong (RCC).

**Figure 6 animals-11-02381-f006:**
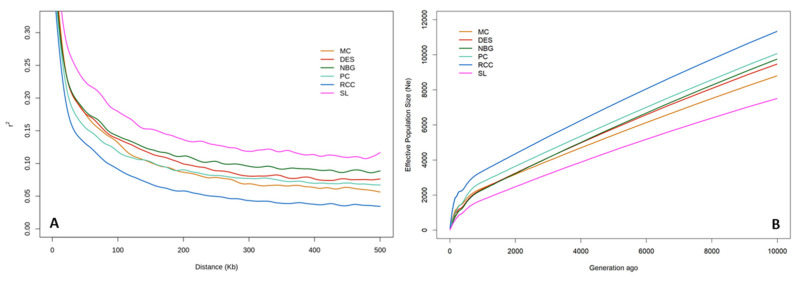
Linkage disequilibrium (LD) decay and effective population size (Ne) of six Bangladeshi cattle populations. LD decay was measured as the average squared correlation coefficient of allele frequencies at pair of loci (r^2^) over genomic distance ranging from 0 to 500 Kb (**A**). Ne of each population was calculated through the SNP based LD analysis over the last 10,000 generations ago (**B**). Cattle breeds/population abbreviations are similar to Figure 1.

**Table 1 animals-11-02381-t001:** Genetic diversity indices for the studied six cattle populations of Bangladesh.

Breed/Population	N	P_N_	MAF ≤ 0.05	Ho ± SD	He ± SD
Red Chittagong (RCC)	89	0.905	0.272	0.250 ± 0.180	0.209 ± 0.129
Pabna (PC)	45	0.891	0.242	0.211 ± 0.166	0.176 ± 0.125
Munshiganj (MC)	26	0.734	0.189	0.215 ± 0.182	0.179 ± 0.136
Non-descript Deshi (DES)	22	0.770	0.154	0.221 ± 0.173	0.185 ± 0.129
North Bengal Grey (NBG)	21	0.818	0.205	0.209 ± 0.170	0.175 ± 0.127
Sahiwal (SL)	15	0.668	0.120	0.226 ± 0.177	0.188 ± 0.130
Merged	218	0.798	0.197	0.222 ± 0.175	0.185 ± 0.129

N: Number of samples; P_N_: Proportion of polymorphic SNPs; MAF: Minor allele frequency up to 5.0%; observed (Ho) and expected (He) heterozygosity for the six cattle populations.

**Table 2 animals-11-02381-t002:** Genetic composition of six cattle populations of Bangladesh compared with indicine and taurine breeds using the ADMIXTURE software.

Breed/ Population	*n*	Cluster 1 (Unspecified)	Cluster 2 (SL)	Cluster 3 (BRM)	Cluster 4 (NEL)	Cluster 5 (HOL)	Cluster 6 (JER)
RCC	89	0.39	0.56	0.01	0.02	0.01	0.01
PC	45	0.11	0.83	0.01	0.01	0.03	0.01
MC	26	0.13	0.83	0.00	0.03	0.00	0.00
NBG	22	0.13	0.76	0.03	0.05	0.04	0.00
DES	21	0.13	0.78	0.02	0.05	0.01	0.01
SL	15	0.04	0.86	0.03	0.06	0.01	0.00
BRM	25	0.02	0.11	0.83	0.04	0.00	0.00
NEL	21	0.00	0.01	0.01	0.98	0.00	0.00
HOL	30	0.00	0.00	0.00	0.00	0.98	0.02
JER	24	0.00	0.00	0.00	0.00	0.00	1.00

*n*: Number of samples; RCC: Red Chittagong; PC: Pabna; MC: Munshiganj; NBG: North Bengal Grey; DES: Non-descript Deshi; SL: Sahiwal; BRM: Brahman; NEL: Nellore; HOL: Holstein; and JER: Jersey.

## Data Availability

Not applicable.

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
