# Peer review of "Unraveling the Genetic Diversity and Population Structure of Bangladeshi Indigenous Cattle Populations Using 50K SNP Markers"

_animals, 2021, doi:10.3390/ani11082381_

Round 1
Reviewer 1 Report
General comments:
I find this manuscript very interesting about the genetic diversity and population structure of the Bangladeshi cattle. Some sections require careful editing to condense the lengthy sentences and repeating information, especially between Results vs Discussion. The analytical tools/methods used in this study are common, therefore, excessive details can be summarized. The overall outcome of this study should be simplified about the quantitative relatedness among Bangladeshi cattle and their closely related breeds.
Specific comments:
Title:
Simplify to “Genetic Diversity and Population Structure of Bangladeshi Indigenous Cattle”, given that 50K SNPs as well as some of the analyses are performed on almost 18000 SNPs, so “high-density” is not an appropriate description in the title.
Summary:
Line 18: Please change “invasion” to “imports”.
Line 24-25: This sentence needs attention.
Abstract:
Line 30-33: Very long and convoluted sentence, please break and simplify.
Line 39: “very low proportion of taurine ancestry”. Please quantify.
Lines 40-43: This is not a conclusion.
Line 42: “improvement of their production, resilience”, how?
The abstract requires to focus on key results and major conclusion(s) based on those key results. Please try and avoid the generalized statements.
Introduction:
Line 50: “better reproducibility”, I think the indigenous breeds have generally lower reproductive performance.
Line 51: Delete “withstand ability of”. Moreover, “low maintenance requirements” and “adaptation under low-input management practices” both are the same things.
Line 53: “categorized into five different varieties or types”. Why the authors are reluctant to call them ‘breeds’?
Line 53: Be consistent to use either 3 or 2 letter acronyms for all breeds/populations.
Line 59: “has been” to “were”
Line 68: “risk of extinction”, please provide any reference.
Lines 74-75: Please simplify.
Lines 78-87: This is overselling of SNP50K chip, which started over 15 years ago.
Line 111: “Precautions”, please explain.
Lines 124-125: this sentence is repeated in results (181-182). Should delete from here as the number being reported before outlining the QC criteria.
Lines 156-158: Why the QC has removed almost 67% SNPs from 50K array, any reason? Were the QCs applied to within breed or the combined data?
Lines 166: “NJ”, please write in full at first use.
Lines 171-172: Move the “(r2)” after “correlation coefficient”.
Line 176: “R program”, any specific package? Please also provide the reference to “R program”.
Line 209: Insert space after “PC1”
Lines 219-220: “All individuals dispersedly distributed without formation of any specific cluster”, please reword and also not clear which population is being referred here.
Lines 231-233: Delete as it sounds like a figure caption.
Line 419: “small sample size”, it is one of the factors potentially confounding this study, and most of the previously done indicine related work. In addition, authors should also discuss the limited power of using relatively lower density of SNPs.
Reviewer 2 Report
In the manuscript entitled“Unraveling the Genetic Diversity and Population Structure of Bangladeshi Indigenous Cattle Populations Using High-Density SNP Markers”(animals-1318079 . In this study, the author detected the genome information of 6 native Bangladeshi cattle populations using the Illumina Bovine SNP50K BeadChip, combined with data from 19 other breeds around the world, to analyze the genetic diversity, population structure, population differentiation, and linkage disequilibrium, which provides a foundation for future breeding and improvement of information. The study is of interest and convincing. This manuscript is suitable to be considered for publication but some details still need to be improved. The following are some questions and suggestions for modification:
- These six breeds of cattle are unfamiliar to other readers, and the author might consider creating a simple distribution map
- It is recommended to add a percent sign to PC1 and PC2 in the upper right corner of Figure 2.
- How are the ratios of genetic composition in Table 2 calculated? Is it the average of the values when K =6 ?
- The species names at the bottom of Figure 4 have been changed to horizontal for easier reading.
- It is suggested that the branches of different breeds on the phylogenetic tree in Figure 5 be colored so that the differentiation of these species can be clearly seen.
- The author does not seem to mention the limitations of the study in the discussion. Are the chips used in this study applicable to zebu breeds? Will the bias be greater? Does the small number of SNP markers detected based on this low density chip have any influence on the results?
- There is a strong gene flow among the Bengal cattle, but the difference between them is large, such as hair color and shape, and it is suspected that the expression is improper.
- In this paper, the data of PCA is complex, so it is suggested to simplify it
- Our findings provide a comprehensive genomic information on indigenous cattle populations of Bangladesh that could be utilized in their future conservation and genetic improvement programs. This sentence is not properly expressed.
